# Effect of intentional restriction of venous return on tissue oxygenation in a porcine model of acute limb ischemia

Wonho Kim[1,2]* , Donghoon Choi[3], Yangsoo Jang[3], Chung Mo Nam[4], Seung-Ho Hur[5], Myeong-Ki Hong[3]*

**1** Department of Internal Medicine, Graduate School, Yonsei University College of Medicine, Seoul, Republic of Korea, **2** Division of Cardiology, Eulji University Hospital, Eulji University School of Medicine, Daejeon, Republic of Korea, **3** Division of Cardiology, Severance Cardiovascular Hospital, Yonsei University Health System, Seoul, Republic of Korea, **4** Department of Preventive Medicine, Yonsei University College of Medicine, Seoul, Republic of Korea, **5** Division of Cardiology, Department of Internal Medicine, Keimyung University Dongsan Medical Center, Daegu, Republic of Korea

☯ These authors contributed equally to this work.
* cardiokwh@gmail.com (WK); mkhong61@yuhs.ac (M-KH)

## Abstract

### Introduction

A sufficient oxygen supply to ischemic limb tissue is the most important requirement for wound healing and limb salvage. We investigated whether partial venous occlusion in the common iliac vein (CIV) causes a further increase of venous oxygenation in a porcine model of acute hindlimb ischemia.

### Materials and methods

In 7 pigs, the model of acute hindlimb ischemia was created with intra-vascular embolization of the common iliac artery (CIA). The arterial and venous oxygen saturation was evaluated at different moments. Oxygen saturation was evaluated at baseline (T0), just after the arterial embolization (T1), at 10 minutes (T2), at 20 minutes (T3), and at 40 minutes (T4). Next, an intentional partial venous occlusion was achieved by inflating the vascular balloon at the level of the right CIV. Then, blood sampling was repeated at 5 minutes (T5), at 15 minutes (T6), and at 25 minutes (T7).

### Results

The arterial oxygen saturation in the right SFA was similar during all phases. In contrast, after arterial embolization, an immediate reduction of venous oxygen saturation was observed (from 85.57 ± 1.72 at T0 to 71.86 ± 7.58 at T4). After the partial venous occlusion, interestingly, the venous oxygen saturations (T5-T7) were significantly increased, again. The venous oxygen saturations evaluated in the hindlimb ischemia with partial venous occlusion and in the control limb (without partial venous occlusion) were significantly over time. Venous oxygen saturations in the experimental limbs were higher than those in the

**Data Availability Statement:** All relevant data are within the manuscript and its Supporting Information files.

**Funding:** The authors received no specific funding for this work.

**Competing interests:** The authors have declared that no competing interests exist.

control limbs (79.28 ± 4.82 vs 59.00 ± 2.82, p-value <0.001, 79.71 ± 4.78 vs 60.00 ± 4.24 at T7, p-value <0.001).

## Conclusions

Partial venous occlusion results in an increase of venous oxygen saturation in the ischemic limb, while significant changes in venous oxygen saturation are not observed in the control limb. An explanation for this may be that the oxygen consumption in the limb tissue is increased because it gets congested with the partial venous occlusion in the right CIV.

## Introduction

Peripheral artery disease (PAD) is a cause of significant morbidity, mortality, and disability due to limb loss [1]. In particular, acute limb ischemia (ALI) occurs when there is a sudden decrease in limb perfusion that threatens the viability of the limb and requires urgent treatment to prevent loss of limb [2]. A lack of oxygen due to an interruption of the blood supply to an acutely occluded limb causes the accumulation of various metabolites, production of reactive oxygen species (ROS) and an inflammatory reaction in conjugation with tissue swelling, threatening the possibility of limb loss and even death [3]. Subsequently, a sufficient oxygen supply to the ischemic limb tissue is the most important requirement for the promotion of wound healing and limb salvage. For limb salvage and restoring ambulatory function, therapies targeting ALI are directed primarily towards mechanical revascularization with an endovascular or surgical approach because essentially no effective medical therapies exist. Unfortunately, revascularization is often of limited benefit, and the clinical outcomes of the revascularization are not always satisfactory [2,4].

Interestingly, previous investigators showed that an intentional narrowing of the coronary sinus lumen elevates the coronary venous pressure, leading to an increase in capillary and arteriolar dilatation, a lower resistance to blood flow, and the restoration of the normal endocardial/epicardial blood flow ratio in patients with refractory angina [5,6]. As a consequence of the enhanced blood flow to the ischemic myocardium, this process increases the oxygen supply and its metabolism in ischemic myocardium.

In this study, a porcine model of acute hindlimb ischemia was created by intra-vascular embolization of the common iliac artery (CIA). By using models, we investigated if partial venous occlusion in the common iliac vein (CIV) might cause a further increase of venous oxygenation.

## Materials and methods

This study was approved by the Yonsei University Institutional Animal Care and Use Committee. Sexually mature female swine (N = 8, 6 to 8 months of age) were obtained from a local vendor. The animals used in this study were miniature pigs originated from Korean Jeju island native (Cronex Co., Ltd., Osong, Republic of Korea). All the animals received humane care in compliance with the Animal Welfare Act and "The Guide for the Care and Use of Laboratory Animals" formulated by the Institute of Laboratory Animal Research [7]. The animals were housed individually following standard laboratory conditions (temperature: 19–25 ˚C; humidity: 30–70%; ventilation: 10–15 per hour; light: 150–300 Lux; light cycle: twice per day (8 AM-8PM); nose: 45 dB) and fed a standard laboratory pellet diet and water ad libitum. A veterinarian dedicated to our laboratory supervised and helped with the study. At the end of this study,

animals were euthanized, according to the American Veterinary Medical Association guidelines for the euthanasia of animals. Briefly, under deep anesthesia, the animals received bolus injections of potassium chloride.

## Procedures

Anesthesia was induced by using intra-muscular injections of ketamine (20 mg/kg) and xylazine (2 mg/kg). After adequate systemic anesthesia was attained, the animals were placed in the supine position under mechanical ventilation, and isoflurane (1–2%) was delivered with a precision vaporizer and a circle absorption breathing system with periodic arterial blood gas monitoring. Cardiovascular monitoring was used during the whole procedure to control the heart rate and blood pressure. After surgical exposure of both carotid arteries and one carotid vein under sterile conditions, two 8 Fr arterial sheathes and one 9 Fr vein sheath were inserted (Fig 1). Intra-venous heparin (150 units/kg) was injected to maintain an activated clotting time of $\geq$ 250 seconds. First, a 0.014-inch hydrophilic guidewire was carefully inserted into the left carotid artery and navigated down to the right CIA. An 8 Fr multi-purpose guiding catheter was then advanced over the guidewire into the right CIA. A micro-catheter through the right carotid artery was placed in the right superficial femoral artery (SFA). The role of micro-catheter is to obtain arterial blood sampling in the right SFA when the right CIA is embolized to induce hindlimb ischemia. Second, a 0.014-inch hydrophilic guidewire through the left carotid vein was advanced in the inferior vena cava (IVC) and selectively, down into the right CIV. A 9 Fr multi-purpose guiding catheter was moved down over the guidewire and positioned in the CIV.

## Lower limb ischemia model & blood sampling

Baseline angiography was performed in the right leg (Fig 2). Embolization to induce hindlimb ischemia in the right CIA was performed using an Amplatzer Vascular Plug (AVP; St/ Jude Medical, St. Paul, MN), which acts as an embolic agent by promoting clot formation. To increase the thrombogenicity of the AVP, it was oversized by 30–50% of the estimated diameter of the right CIA. The occlusion time is highly variable depending on the caliber of the vessel, the coagulation status, the flow dynamics, and the size of the AVP. Therefore, repeated angiography was frequently performed to see if complete stasis of the injected contrast was noted. Blood sampling (Fig 3) for evaluating oxygen saturation was obtained from the right SFA and CIV at baseline (T0), just after the arterial occlusion with embolization (T1), at 10 minutes (T2), at 20 minutes (T3), and at 40 minutes (T4). Next, an intentional partial (90%) venous occlusion was achieved by inflating the vascular balloon at the level of the right CIV. Subsequently, blood sampling was repeated 5 minutes (T5) after the partial venous occlusion, at 15 minutes (T6), and at 25 minutes (T7). The left leg in two pigs served as a control limb; the left CIA was embolized while the left CIV was not occluded. In those legs, blood sampling was obtained 40 minutes after the left CIA occlusion in the left CIV with the same method as above (T5-1, T6-1, and T7-1). Blood gas analysis was performed to measure oxygen saturation, carbon dioxide, bicarbonate, and acid-base balance (pH). A proper blood sample for blood gas analysis consists of a 2 to 3 ml blood specimen collected anaerobically in a 3 ml plastic syringe fitted with a small-bore needle. Heparin was added to the syringe as an anticoagulant. Any air bubbles inadvertently introduced during sampling were promptly evacuated. Hypoxic damage by arterial embolization was defined as T4 minus T0 in vein saturation, and treatment effect with partial venous occlusion was defined as T7 minus T4 in vein saturation, respectively. Mean venous pressure was measured during the study period. All the mean venous pressure values were calculated over a stable period.

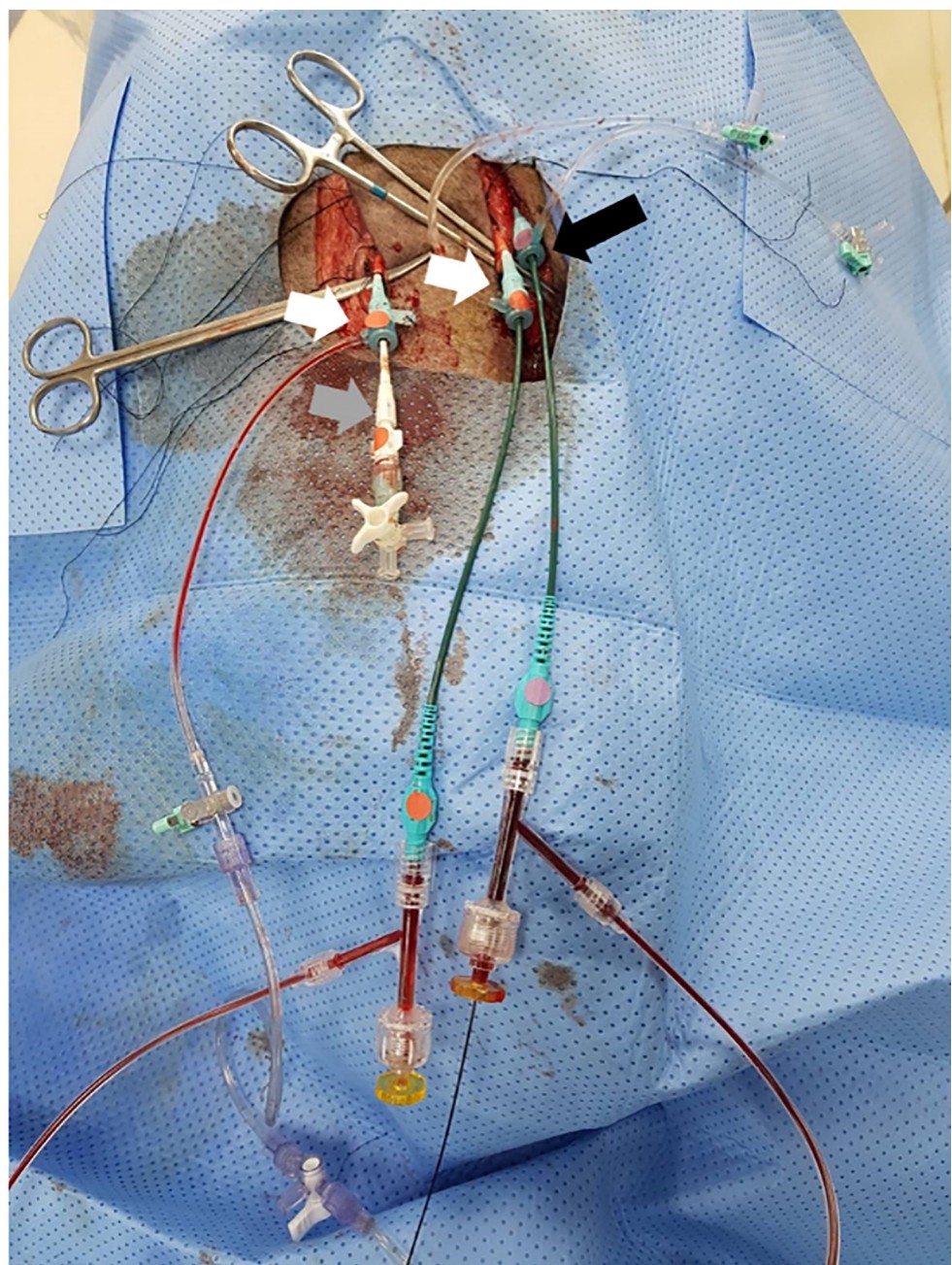

**Fig 1. Preparation of an animal model of ALI.** After surgical exposure of both carotid arteries (white arrows) and one carotid vein (black arrow) under sterile conditions, two 8 Fr arterial sheaths (white arrows) and one 9 Fr vein sheath (black arrow) were inserted. Note that a micro-catheter (grey arrow) is inserted in the right carotid artery, an 8 Fr multi-purpose guiding catheter in the left carotid artery, and a 9 Fr multi-purpose guiding catheter in the left carotid vein. Abbreviations: ALI, acute limb ischemia.

## Statistical analysis

Dichotomous data were reported as a number and percentage. Continuous data were reported as the mean and the standard deviation. Student t test was used to analyze continuous data and a chi-square test for dichotomous data. Arterial and venous oxygen saturations at different momentums were submitted to a repeated measures analysis of variance (ANOVA). All tests

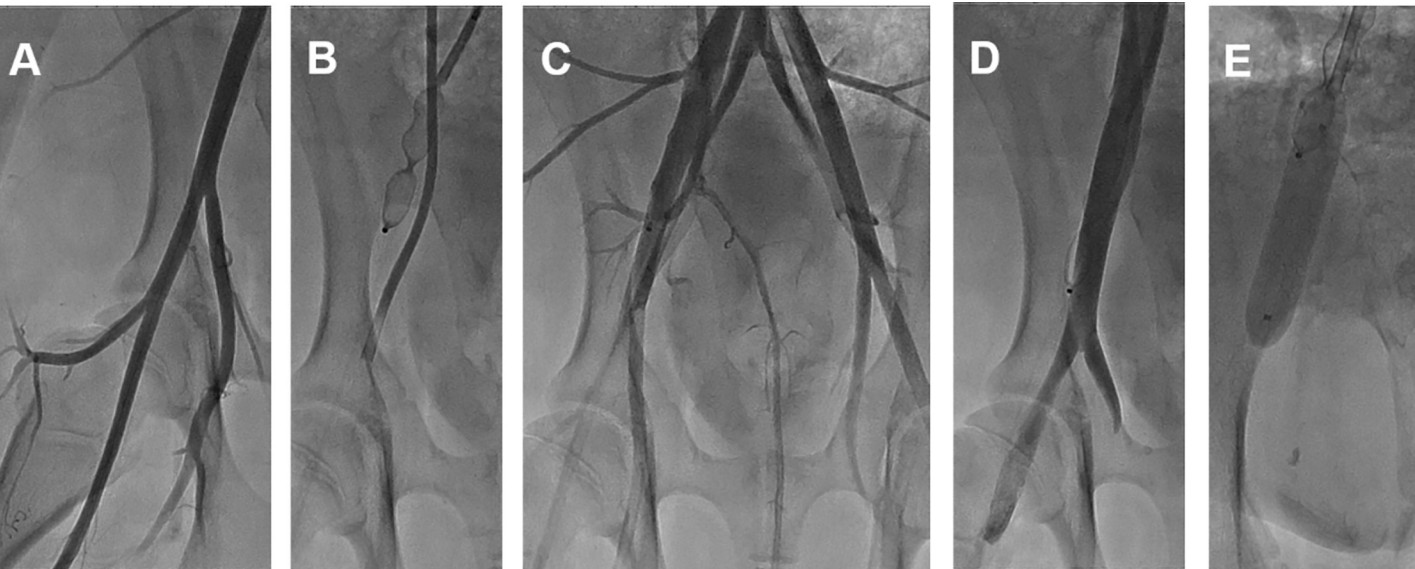

**Fig 2. Representative images of the porcine model of ALI.** The pigs underwent intravascular induction of ALI. Baseline angiography (A), Embolization of the CIA. ALI was generated by the deployment of an Amplatzer Vascular Plug II in the CIA, followed by an angiography to verify occlusion (B), representative angiography after CIA occlusion (C), Baseline venography (D), Venous blood flow restriction with partial balloon inflation (E). Abbreviations: ALI, acute limb ischemia; CIA, common iliac artery.

were two-sided, and statistical significance was defined as a p-value of less than 0.05. The analyses were performed using the SPSS version 20 software (SPSS Inc., Chicago, Illinois).

## Results

The model of acute hindlimb ischemia was successfully created in all 8 pigs. One pig died of sudden cardiac death due to iatrogenic catheter induced ventricular fibrillation. The seven surviving pigs developed acute hindlimb ischemia. The arterial oxygen saturation in the right SFA was similar during all phases. There were no significant differences between the values observed at T1-T4 and T5-T7 (Fig 4A and Table 1). In contrast, after arterial embolization, an immediate reduction of venous oxygen saturation was observed (from $85.57 \pm 1.72$ at T0 to $71.86 \pm 7.58$ at T4). After partial venous occlusion, interestingly, the venous oxygen saturations (T5-T7) were significantly increased, again. Repeated measures ANOVA show a significant difference after the partial venous occlusion concerning venous oxygen saturation (p-value < 0.001) (Fig 4B and Table 1). The venous oxygen saturations evaluated in the hindlimb ischemia with partial venous occlusion and in the control limb (without partial venous occlusion) were significant over time. Venous oxygen saturations in the experimental limb were higher than those in the control limb ($79.28 \pm 4.82$ vs $59.00 \pm 2.82$, p-value <0.001, $79.71 \pm 4.78$ vs $60.00 \pm 4.24$ at T7, p-value <0.001) (Table 2). The largest changes in venous oxygen saturation during partial venous occlusion were observed at T6 (from $74.14 \pm 9.56$ at T5 to $79.28 \pm 4.82$ at T6). Overall, venous oxygenation was significantly increased after partial venous occlusion (from $74.14 \pm 9.56$ at T5 to $79.71 \pm 4.78$ at T7, p-value <0.001) (Fig 4C), while its largest change was observed at T6 (from $74.14 \pm 9.56$ at T5 to $79.28 \pm 4.82$ at T6). Fig 5 shows a representative image of the mean venous blood pressure response during the study period. A significant decline after arterial embolization (from $18.9 \pm 1.3$ at T0 to $3.0 \pm 1.2$ mmHg at T1, p-value <0.001) and elevation were observed (from $3.0 \pm 1.2$ at T0 to $7.1 \pm 1.3$ mmHg at T1, p-value <0.001) in the mean venous pressure. Fig 6 shows that hypoxic damage

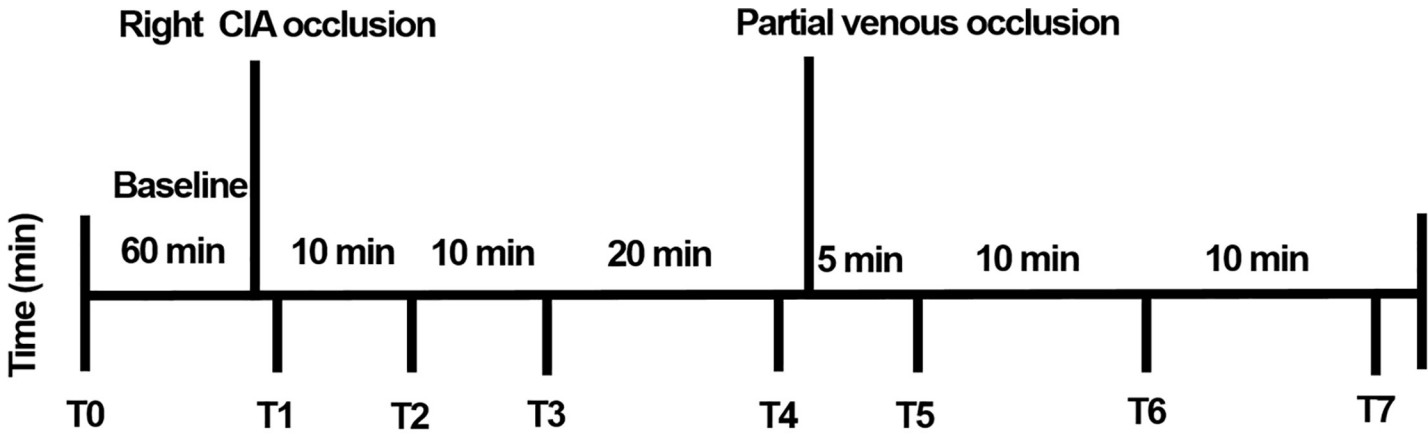

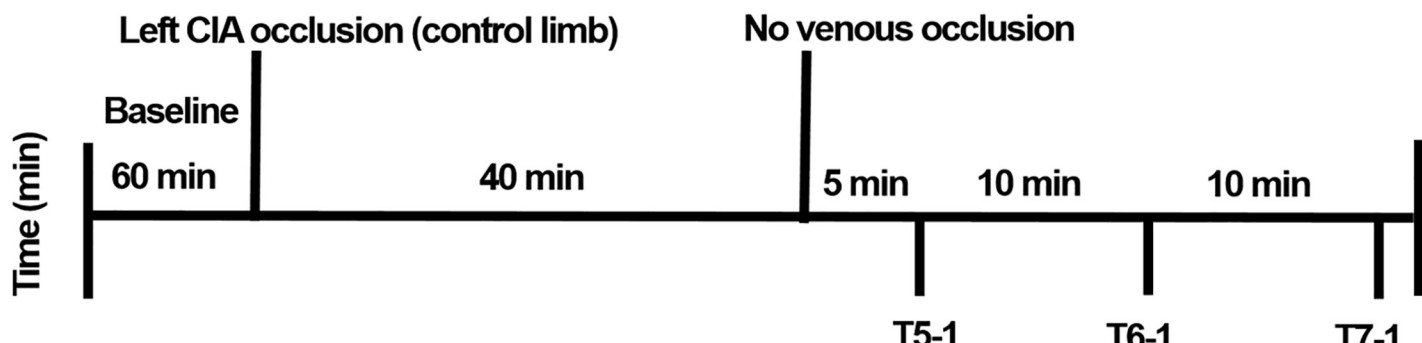

**Fig 3. Experimental study protocol and blood sampling for evaluating oxygen saturation (right SFA and the right CIV).** Blood sampling for evaluating oxygen saturation was obtained from the right SFA and the right CIV at baseline (T0), just after the arterial occlusion with embolization (T1), at 10 minutes (T2), at 20 minutes (T3), and at 40 minutes (T4). Next, an intentional partial (90%) venous occlusion was achieved by inflating the vascular balloon at the level of the right CIV. Subsequently, blood sampling was repeated at 5 minutes (T5) after the partial venous occlusion, at 15 minutes (T6), and at 25 minutes (T7). The left leg in two pigs served as a control limb; the left CIA was embolized while the left CIV was not occluded. In those legs, blood sampling was obtained 40 minutes after the left CIA occlusion in the left CIV with the same method as above (T5-1, T6-1, and T7-1). Abbreviations: CIA, common iliac artery; SFA, superficial femoral artery; CIV, common iliac vein.

by arterial embolization is positively associated with treatment effect with venous partial occlusion (r = 0.788, P-value = 0.035). This strong positive correlation might be interpreted that the more severe hypoxic damage, the greater would be the treatment effect with partial venous occlusion. On the other hand, we did not detect a significant change in the level of carbon dioxide, bicarbonate or pH during the study period.

## Discussion

In this porcine model of acute hindlimb ischemia, arterial embolization decreases venous oxygen saturation over time. However, intentional partial venous occlusion results in an increase of venous oxygen saturation in the ischemic limb, while significant changes in venous oxygen saturation are not observed in the control limb. An explanation for this may be that the oxygen consumption in the limb tissue is increased because it gets congested with the partial venous occlusion in the right CIV. Long term effects of the partial venous occlusion are beyond the spectrum of this study. However, our result may lead to investigating an optional therapy

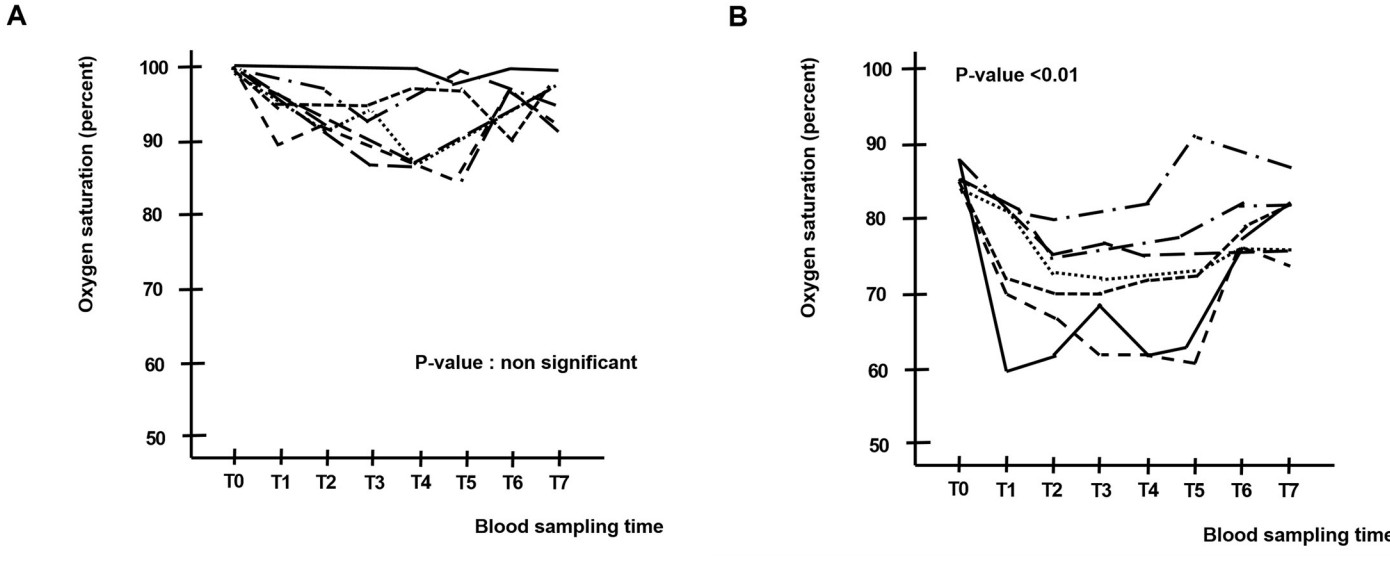

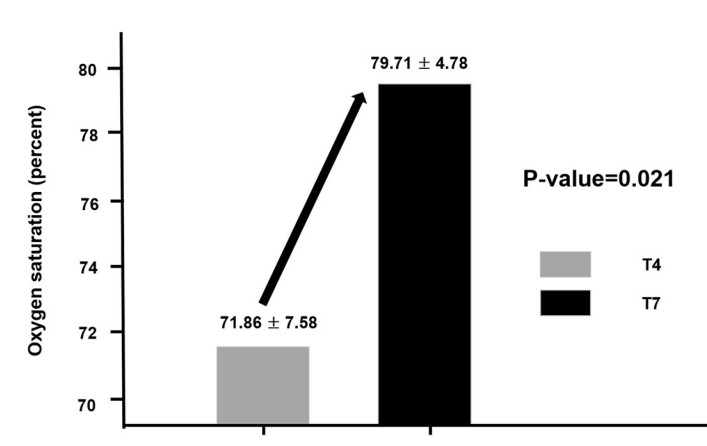

**Fig 4.** Blood sampling of the right SFA (A) and right CIV (B) at different moments (From T0 to T7). Effect of partial venous occlusion on venous oxygen saturation over time (from T4 to T7) (C). Abbreviations: SFA, supercifial femoral artery; CIV, common iliac vein.

**Table 1. Blood sampling of the SFA and CIV at different moments (From T0 to T7).**

| SFA (percent) | | CIV (percent) | |
|---|---|---|---|
| T0 | 100 | T0 | 85.57 ± 1.72 |
| T1 | 98.43 ± 1.27 | T1 | 75.43 ± 8.44 |
| T2 | 98.29 ± 1.11 | T2 | 71.71 ± 5.93 |
| T3 | 97.57 ± 1.51 | T3 | 72.57 ± 6.35 |
| T4 | 97.71 ± 1.97 | T4 | 71.86 ± 7.58 |
| T5 | 98.29 ± 2.13 | T5 | 74.14 ± 9.56 |
| T6 | 98.29 ± 1.60 | T6 | 79.29 ± 4.82 |
| T7 | 98.57 ± 0.97 | T7 | 79.71 ± 4.78 |

Values are n (%) or mean ± SD, as appropriate. Abbreviations: SFA, supercifial femoral artery; CIV, common iliac vein.

**Table 2. Comparision of the venous oxygen saturations between the experimental limbs with arterial embolization and the control limbs.**

|  | Experimental limb | Control limb | p-value |
|---|---|---|---|
| T5 | 74.14 ± 9.56 | 57.50 ± 7.77 | 0.061 |
| T6 | 79.28 ± 4.82 | 59.00 ± 2.82 | < 0.001 |
| T7 | 79.71 ± 4.78 | 60.00 ± 4.24 | < 0.001 |

Values are n (%) or mean ± SD, as appropriate.

targeted at increasing tissue oxygenation, which could be used alone or in combination with mechanical revascularization to improve clinical care.

Mechanical revascularization is a fundamental strategy for limb preservation. However, this strategy is occasionally unsuccessful at improving limb salvage. Investigational approaches derived from animal models or preclinical studies, like stem cell-based or gene therapies, have also been largely ineffective [8,9]. Because there is still much debate on the best clinical and therapeutic strategies, it remains unclear just how good is good enough when it comes to

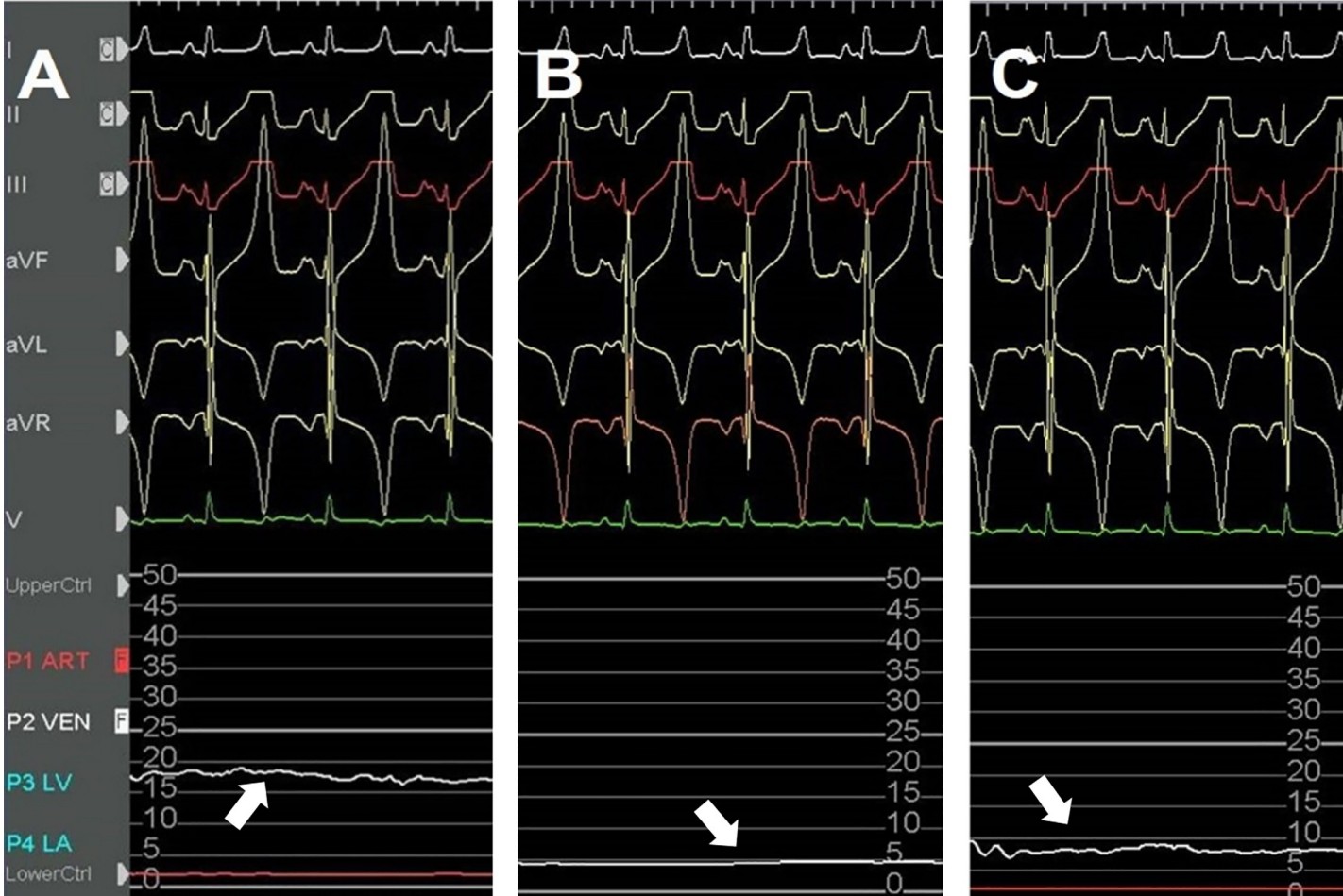

**Fig 5. Representative image of the venous blood pressure response in the ischemic limb.** White arrows show a recording of the mean venous pressure, at baseline (T0), just after the arterial occlusion (T1), and 25 minutes after partial vein occlusion (T7). A decline after arterial occlusion (B) and an elevation after partial vein occlusion were observed in the mean venous pressure (C).

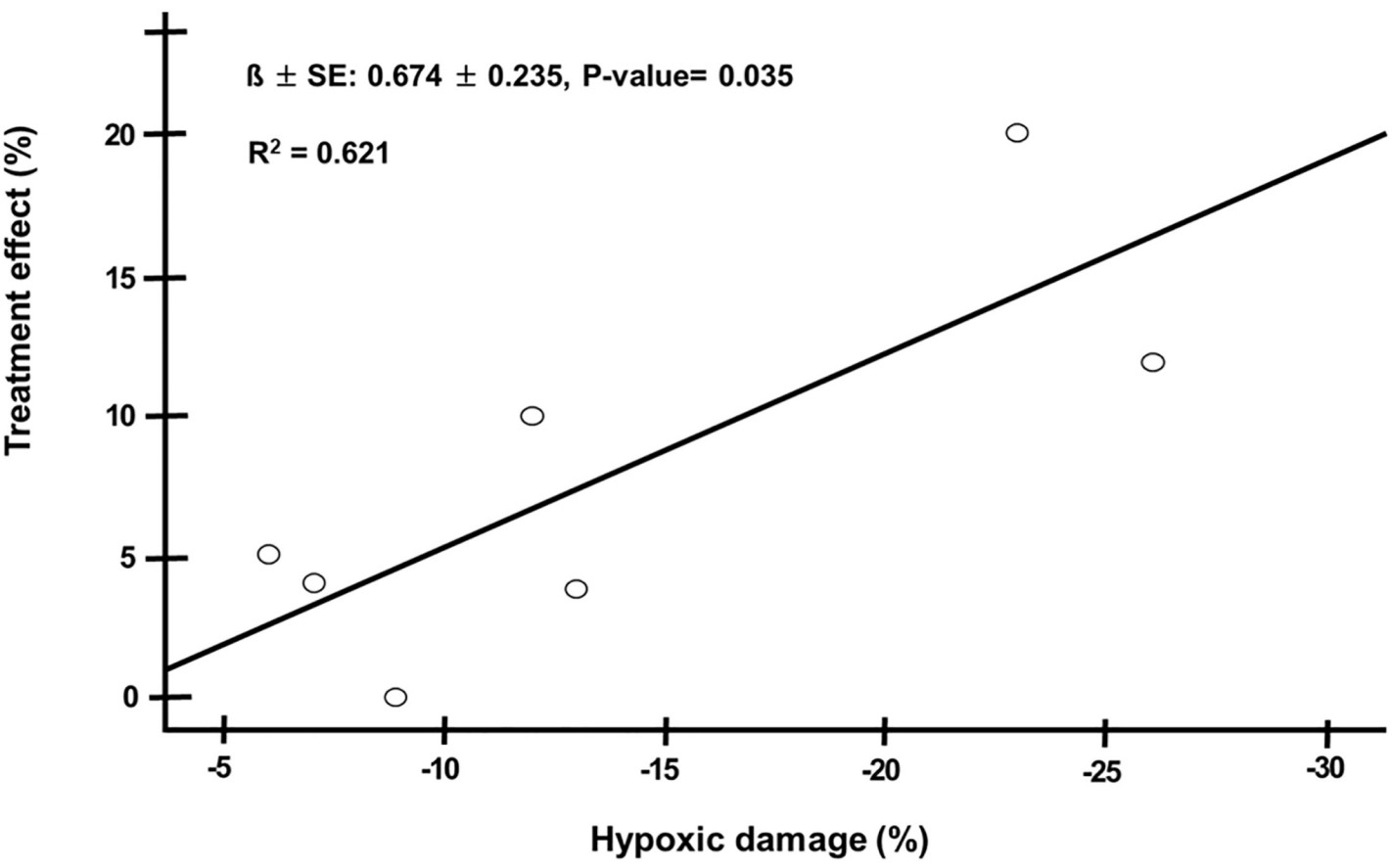

**Fig 6. Spearman correlation was performed for bivariate correlations between hypoxic damage by arterial embolization (defined as T4 minus T0 in vein saturation) and treatment effect with partial venous occlusion (defined as T7 minus T4 in vein saturation) to establish a relationship.**

increasing the blood supply to an ischemic limb. In clinical scenarios, there are patients for whom the revascularization method is not feasible for them. The intentional partial venous restriction in the CIV for oxygenation in the ischemic limb may assist in wound healing and decrease the risk of major amputation.

Our experimental idea got inspiration from Beck's study, which demonstrated in a dog model of acute coronary artery ligation that the partial narrowing of the coronary sinus is associated with a reduction in the infarct size and mortality by a doubled degree of retrograde backflow from the distal stump of the occluded coronary artery [10]. Coronary sinus narrowing leads to the development of an upstream pressure gradient that results in the redistribution of blood from the less ischemic epicardium to the ischemic endocardium [11].

An increase in oxygenated venous blood might be seen in the ischemic limb with both partial and complete venous occlusion. Enriched oxygen in tissue is considered beneficial to the metabolism in the muscle. The obstruction of arterial blood flow causes impaired oxygen delivery, tissue hypoxia and leads to the dysfunction of the electron transport chain in mitochondria. That induces anaerobic metabolism in cells, and the retention of lactic acid may lead to metabolic acidosis. Because tissue oxygen extraction/utilization is impaired in the ischemic limb, an increase in oxygen consumption in the ischemic tissue might not decrease the venous oxygen saturation [12]. Partial venous occlusion in the CIV induces local stasis of highly

oxygenated blood in the ischemic area. It causes a slight venous pressure elevation in the vein (Fig 5), and the sustaining high oxygen level would be helpful for oxygen uptake by ischemic limb tissue, restoring cell metabolism and limb perfusion.

A total venous occlusion may develop a fatal deep vein thrombosis (DVT); there is a risk that a venous thrombosis starts as a partial venous occlusion and progresses over time. In animal experiments, the methods of creating a DVT model are mainly based on flow impairment induced by surgical ligation, endovascular balloon occlusion, or the assistance of specifically designed endovascular devices [13]. However, venous stasis alone, unless prolonged, does not consistently induce venous thrombosis in an intact vein [14]. Nevertheless, we often check if venous blood flow is patent during balloon occlusion to avoid a total venous thrombosis. On the other hand, there is concern that a venous ulcer or venous stasis related life-threatening condition may occur due to venous congestion by an intentional partial venous occlusion, causing pressure in the veins to increase. However, those catastrophic events are unlikely to develop because the intentional partial venous occlusion would be applied, temporarily.

The majority of studies evaluating susceptibility to tissue necrosis in hindlimb ischemia have focused on vascular contributions, such as angiogenesis and collateral artery growth [15,16]. In those studies, limb salvage is logically assumed to be dependent on perfusion and subsequent oxygen delivery. A critical component of limb tissue regeneration after ischemic injury is a sufficient oxygen concentration. On the other hand, oxygen has been used as a target of non-invasive measurements of tissue perfusion in limb ischemia. In clinical practice, the transcutaneous measurement of oxygen partial pressure is a good tool to assess the perfusion status of the foot in limb ischemia [17]. One preliminary study showed that hyperbaric oxygen therapy has anti-microbial and neovascularization effects in patients with successful endovascular revascularization for the affected limb [18].

The most common model of PAD is the hindlimb ischemia model, which is most frequently performed in small animals like rodents and rabbits [19,20]. Generally, the model entails surgical ligation of the femoral artery and its intervening side-branches followed by excision of the vessel, resulting in occlusion of the blood flow and induction of ALI. One might argue that the most important weakness in this study is to use a model of ALI. A criticism of this model is that ALI does not represent the natural history of the most common PAD with gradual development of progressive atherosclerotic lesions. A chronic limb ischemic model with ischemic changes similar to those seen in patients must be ideal, but this work needs several weeks after induction of hindlimb ischemia. They are logistically more complex to handle and not cost effective [21]. Compared to small animals, pigs have a size and physiology that more closely mimic humans.

Meanwhile, previous studies with the porcine hindlimb ischemic model have been limited by rapid collateralization [22,23]. Surgical CIA ligation would completely exclude the possibility of collateralization to the ischemic hindlimb and make a more durable and reproducible ischemic state in the porcine model [21]. However, the human peripheral limb vascular system has numerous collaterals and inter-individual disparities. Generally, irreversible tissue damage in ALI usually occurs within 6 hours after complete arterial occlusion. Although endovascular CIA embolization leads to a stronger reduction in the total amount of blood flow into the ipsilateral limb, it would seem barely feasible to occlude all collateral branches at the initial stage of ALI. Tissue blood flow and perfusion to the skin or limb muscles are significantly decreased, whereas intraluminal arterial saturation in the occluded artery may not be decreased and still has highly saturated blood. In addition, arterial anatomy of the porcine hind limb is different from that in human peripheral limb. In the pig, the distal abdominal aorta generally trifurcates into the right and left iliac arteries, and the common internal iliac trunk which then bifurcates into the right and left internal iliac arteries [24]. Development of collateral vasculature via those internal iliac

arteries might be one of the important mechanisms in compensating for arterial occlusion in the porcine limb ischemic model. Our model was induced with arterial embolization in the CIA, not at the level of distal abdominal aorta. Thus, the patent common internal iliac trunk might facilitate the development of collateral blood flow into the ischemic limb. However, we did not actually show the limb perfusion through collaterals, which could be assessed with peripheral angiography, non-invasive imaging modalities, or histological staining of the ischemic tissues. Swine models fulfilling all ALI characteristics, with evaluating the adequacy of limb perfusion through collaterals intraprocedurally, seem to be lacking in the scientific literature. When it comes to further researches following this study, a major challenge may be having technologies available that can show numerous collaterals in the ischemic limb.

Abrupt total occlusion of the CIA, like surgical CIA ligation, is a rare medical scenario and is hardly seen in PAD patients clinically. Considering the mentioned points, our model is likely to show ischemic changes similar to those seen in patients with ALI. We focused on a straightforward reliable and reproducible endovascular method. Moreover, our method is well equipped for performing our study which is designed to investigate the effect of partial venous occlusion on the oxygenation in porcine hindlimb ischemia.

It is important to recognize several other limitations of this study. First, our model is based on the non-atherosclerotic and normo-lipidemic hindlimb ischemia model. Because the model is done in a swine without any cardiovascular risk factors or comorbidities, the induction of hindlimb ischemia will never perfectly replicate the pathophysiology of clinical PAD. Although the diseased porcine models provide more similar biological responses, the normo-lipemic porcine artery is still recommended as the choice of the porcine model to evaluate pathophysiologic responses in consensus documents [21,25]. This study aimed to investigate the effect of partial venous occlusion on oxygenation in porcine hindlimb ischemia. The main target is not atherosclerotic arteries, but the vein. The presence of cardiovascular risk factors or the onset of ischemia may not influence our expected study outcomes. Second, there are concerns about the small number of experimental animals and a relatively short period of observation time. With the current amount of observations, however, there was an obvious trend towards improvement of the venous oxygen saturation over time, following intentional partial venous occlusion in the CIV. Possibly, the inclusion of an additional time point, e.g. 6 hours after the partial venous occlusion, may have revealed more information about the effect of the intentional restriction of venous return on tissue oxygenation in the ischemic limb. Third, partial venous occlusion was created with percutaneous balloon inflation, causing a 90% stenosis artificially in the CIV by visual estimation. The designated percutaneous device, like a coronary sinus reducer, may provide a more accurate way in which to achieve a stable and durable venous blood flow. However, those devices are not yet available, and it is not likely to be cost effective. Future investigators should clarify the question concerning the designated vein narrowing device. In our opinion, our porcine model with percutaneous balloon inflation could prove suitable for evaluating the effect of partial venous occlusion on venous oxygenation. Finally, our study was insufficient to demonstrate an effect of partial venous occlusion on the inflammatory marker, oxidative stress, and acid-base balance in the ischemic limb. However, this study was not designed or powered to look at the impact on those parameters, although those are likely of great interest. The study period may not have been long to detect significant changes.

## Conclusions

Given the limited evidence-based research to date, the clinical efficacy of using a non-revascularization strategy in limb ischemia is still questionable. Our results suggest a potential positive

therapeutic effect of partial venous occlusion to induce the enhancement of oxygen uptake in the ischemic limb tissue. The effect of a partial venous occlusion on tissue oxygenation in an ischemic limb may deserve to be investigated in patients for whom revascularization therapy is unsuccessful.

## Supporting information

**S1 Table. Blood sampling of the SFA and CIV at different moments (From T0 to T7).** (PDF)

**S2 Table. Comparision of the venous oxygen saturations between the experimental limbs with arterial embolization and the control limbs.** (PDF)

**S1 File. Analysis file.** (XLS)

## Author Contributions

**Data curation:** Wonho Kim.

**Formal analysis:** Wonho Kim.

**Investigation:** Wonho Kim, Donghoon Choi, Yangsoo Jang, Chung Mo Nam, Seung-Ho Hur, Myeong-Ki Hong.

**Validation:** Myeong-Ki Hong.

**Writing – original draft:** Wonho Kim.

**Writing – review & editing:** Wonho Kim.

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
