## [Decision Letter · Decision Letter 0]

21 Aug 2020

PONE-D-20-19360

Effect of intentional restriction of venous return on tissue oxygenation in a porcine model of acute limb ischemia

PLOS ONE

Dear Dr. Kim,

Thank you for submitting your manuscript to PLOS ONE. After careful consideration, we feel that it has merit but does not fully meet PLOS ONE’s publication criteria as it currently stands. Therefore, we invite you to submit a revised version of the manuscript that addresses the points raised during the review process.

We look forward to receiving your revised manuscript.

Kind regards,

Yoshihiro Fukumoto

Academic Editor

PLOS ONE

Journal Requirements:

2.At this time, we request that you  please report additional details in your Methods section regarding animal care, as per our editorial guidelines:

(1) Please state the source of the pigs used in the study  

(2) Please provide details of animal welfare (e.g., shelter, food, water, environmental enrichment)

(3) Please include the method of euthanasia

(4) Please describe the post-operative care received by the animals, including the frequency of monitoring and the criteria used to assess animal health and well-being.  

Thank you for your attention to these requests.

3.Thank you for stating the following financial disclosure:

 [The funders had no role in study design, data collection and analysis, decision to publish, or preparation of the manuscript.].

4. Please amend your manuscript to include your abstract after the title page.

Reviewers' comments:

Reviewer's Responses to Questions

**Comments to the Author**

1. Is the manuscript technically sound, and do the data support the conclusions?

Reviewer #1: No

Reviewer #2: Yes

2. Has the statistical analysis been performed appropriately and rigorously? 

Reviewer #1: I Don't Know

Reviewer #2: I Don't Know

3. Have the authors made all data underlying the findings in their manuscript fully available?

Reviewer #1: No

Reviewer #2: Yes

4. Is the manuscript presented in an intelligible fashion and written in standard English?

Reviewer #1: Yes

Reviewer #2: Yes

5. Review Comments to the Author

Reviewer #1: Kim et al. have reported that an intentional partial venous occlusion in the common iliac vein (CIV) of porcine with acute hindlimb ischemia created by embolization of the ipsilateral common iliac artery (CIA) increased the venous oxygen saturation. The report is interesting; however, there are several issues to be resolved for the next review.

1. There is no abstract in this manuscript.

2. Although the CIA occlusion remained for 65 minutes in total, why did not the arterial oxygen saturation in the ipsilateral superficial femoral artery (SFA) decrease?

3. The authors hypothesized that the intentional partial occlusion of the CIV congested the whole vein, subsequently increased the oxygen consumption in the ischemic limb tissue, and thereby increased the venous oxygen saturation. However, I cannot understand the hypothesis because an increase in the oxygen consumption in the ischemic tissue should further decrease the venous oxygen saturation. Could you clear the doubts away?

4. The authors showed an increase in venous oxygen saturation after the intentional partial occlusion of the CIV. However, the authors should additionally test whether the CIV occlusion could increase arterial oxygen saturation and tissue oxygenation in the peripheral arteries of the occluded CIA.

5. Please show if the intentional partial occlusion of the CIV can reduce ischemic damage (e.g. necrosis, inflammatory, oxidative stress, and apoptosis) in the ischemic limb.

6. The authors should additionally test the effect of the intentional partial occlusion of the CIV in porcine with chronic hindlimb ischemia, which is created by the CIA occlusion for several days. If impossible, the authors should not describe a clinical perspective regarding the CIV occlusion method for treating patients with critical limb ischemia.

7. As for the statistical analysis of changes in the oxygen saturation following the intentional partial occlusion of the CIV, was the analysis adjusted by the baseline oxygen saturation before the occlusion?

Reviewer #2: The authors present a single institution study looking at the effect of intentional restriction of venous return on tissue oxygenation in a porcine model of acute limb ischemia and investigate whether partial venous occlusion in the common iliac vein (CIV) causes a further increase of venous oxygenation in a porcine model of acute hindlimb ischemia.

. Just unter 10 porcines were analyzed. While this is interesting work, I have some questions and suggestions for revisions:

I think the number of the study is to small to draw conculsion, the authors should soften their statements

In Discussion part line 15 “..An expectation for this may be that the oxygen consumption in the limb tissue is increased..” do you mean expectation or explanation . I think you mean explanation.

The pathophysiological reasons of venous occlusion and increased oxygen saturation is not becoming clear. Please discuss it better. How do you explain this inthe peripheral system?

We know in the peripheral system a venous congestion results in trophic skin disorder and venus ulcera and is contraindicated for wound healing.

How do you measure the saturation, this should be clarified.

Page 3 line 21-23 : The authors try to transferr the pathophysiological process in the heart to the perpheral vessels. I think this need to be discussed much deeper.

Arterial occlusion combined with venous occlusion results in the peripheral system to the situation phlegmasia coerlia dolens, which is a life threating condition. Who was lactat, do you measure any labor parameters.

6. PLOS authors have the option to publish the peer review history of their article (what does this mean?). If published, this will include your full peer review and any attached files.

Reviewer #1: No

Reviewer #2: No

---

## [Author Response · Author response to Decision Letter 0]

18 Sep 2020

Journal Requirements:

2.At this time, we request that you please report additional details in your Methods section regarding animal care, as per our editorial guidelines:

(1) Please state the source of the pigs used in the study 

A sentence was added in the method section as below :

The animals used in this study were miniature pigs originated from Korean Jeju island native (Cronex Co., Ltd., Osong, Republic of Korea).

(2) Please provide details of animal welfare (e.g., shelter, food, water, environmental enrichment) : Sentences were added in the method section as below :

The animals were housed individually following standard laboratory conditions (temperature: 19-25 ˚C; humidity: 30-70%; ventilation: 10-15 per hour; light: 150-300 Lux; light cycle: twice per day (8 AM- 8PM); nose: 45 dB) and fed a standard laboratory pellet diet and water ad libitum.

(3) Please include the method of euthanasia & (4) Please describe the post-operative care received by the animals, including the frequency of monitoring and the criteria used to assess animal health and well-being; 

To address this reviewer’s comment, we added some sentences in the method section as below :

At the end of this study, the animals were euthanized, according to the American Veterinary Medical Association guidelines for the euthanasia of animals. Briefly, under the deep anesthesia, the animals received bolus injections of potassium chloride.

3.Thank you for stating the following financial disclosure:

[The funders had no role in study design, data collection and analysis, decision to publish, or preparation of the manuscript.].

Please clarify the sources of funding (financial or material support) for your study. List the grants or organizations that supported your study, including funding received from your institution. State what role the funders took in the study. If the funders had no role in your study, please state: “The funders had no role in study design, data collection and analysis, decision to publish, or preparation of the manuscript.” If any authors received a salary from any of your funders, please state which authors and which funders. If you did not receive any funding for this study, please state: “The authors received no specific funding for this work.”

A) Following the conclusion, we inserted the sentence “The authors received no specific funding for this work” 

A) The amended statements included within the cover letter

4. Please amend your manuscript to include your abstract after the title page.

A) Abstract is included

5. Review Comments to the Author

Reviewer #1: Kim et al. have reported that an intentional partial venous occlusion in the common iliac vein (CIV) of porcine with acute hindlimb ischemia created by embolization of the ipsilateral common iliac artery (CIA) increased the venous oxygen saturation. The report is interesting; however, there are several issues to be resolved for the next review.

1. There is no abstract in this manuscript.

A) Abstract is included

2. Although the CIA occlusion remained for 65 minutes in total, why did not the arterial oxygen saturation in the ipsilateral superficial femoral artery (SFA) decrease?

A) To address this reviewer’s comment, we added some explanations and modified the previous paragraphs in the discussion section as below :

Meanwhile, previous studies with the porcine hindlimb ischemic model have been limited by rapid collateralization [21,22]. Surgical CIA ligation would completely exclude the possibility of collateralization to the ischemic hindlimb and make a more durable and reproducible ischemic state in porcine model [20]. However, human peripheral limb vascular system has numerous collaterals and inter-individual disparities. Generally, irreversible tissue damage in acute limb ischemia usually occurs within 6 hours after complete arterial occlusion. Although endovascular CIA embolization lead to a stronger reduction in the total amount of blood flow into the ipsilateral limb, it would seem barely feasible to occlude all collateral branches at the initial stage of acute limb ischemia. Tissue blood flow and perfusion to the skin or limb muscles are significantly decreased, whereas intraluminal arterial saturation in the occluded artery may not be decreased and still has highly saturated blood. Abrupt total occlusion of the CIA, like surgical CIA ligation, is a rare medical scenario and is hardly seen in PAD patients clinically. Considering the mentioned points, our model is likely to show ischemic changes similar to those seen in patients with ALI. We focused on a straight-forward reliable and reproducible endovascular method. Moreover, our method is well equipped for performing our study which is designed to investigate the effect of partial venous occlusion on the oxygenation in porcine hindlimb ischemia.

3. The authors hypothesized that the intentional partial occlusion of the CIV congested the whole vein, subsequently increased the oxygen consumption in the ischemic limb tissue, and thereby increased the venous oxygen saturation. However, I cannot understand the hypothesis because an increase in the oxygen consumption in the ischemic tissue should further decrease the venous oxygen saturation. Could you clear the doubts away?

A) To address the reviewer comments and support our hypothesis, we add a paragraph in the discussion as below : 

Enriched oxygen in tissue is considered beneficial to the metabolism in the muscle. The obstruction of arterial blood flow causes impaired oxygen delivery, tissue hypoxia and leads to the dysfunction of the electron transport chain in mitochondria. That induces anaerobic metabolism in cells, and the retention of lactic acid may lead to metabolic acidosis. Because tissue oxygen extraction/utilization is impaired in the ischemic limb, an increase in oxygen consumption in the ischemic tissue might not decrease the venous oxygen saturation [12]. Partial venous occlusion in the CIV induces local stasis of highly oxygenated blood in the ischemic area. It causes a slight venous pressure elevation in the vein, and the sustaining high oxygen level would be helpful for oxygen uptake by ischemic limb tissue, restoring cell metabolism and limb perfusion.

A) Figure 5 regarding venous pressure was added in the method (1), result (2) as below : 

(1) Mean venous pressure were measured during the study period. All the mean venous pressure values were calculated over the stable period.

(2). Fig 5 shows a representative image of the mean venous blood pressure response during the study period. A significant decline after arterial embolization (from 18.9 ± 1.3 at T0 to 3.0 ± 1.2 mmHg at T1, p-value <0.001) and an obvious elevation were observed (from 3.0 ± 1.2 at T0 to 7.1 ± 1.3 mmHg at T1, p-value <0.001) in the mean venous pressure. 

4. The authors showed an increase in venous oxygen saturation after the intentional partial occlusion of the CIV. However, the authors should additionally test whether the CIV occlusion could increase arterial oxygen saturation and tissue oxygenation in the peripheral arteries of the occluded CIA.

A) Arterial oxygen saturation was not changed significantly during the study. We mentioned about the aformentioned question, and inserted the explanation in the discussion as below: 

Although endovascular CIA embolization lead to a stronger reduction in the total amount of blood flow into the ipsilateral limb, it would seem barely feasible to occlude all collateral branches at the initial stage of acute limb ischemia. Tissue blood flow and perfusion to skin or limb muscles are significantly decreased, whereas intraluminal arterial saturation in the occluded artery may not be decreased and still has highly saturated blood. 

5. Please show if the intentional partial occlusion of the CIV can reduce ischemic damage (e.g. necrosis, inflammatory, oxidative stress, and apoptosis) in the ischemic limb.

A) We added a sentence in the result :

We did not detect a significant change in the level of carbon dioxide, bicarbonate or pH during the study period.

A) We added a response in the limitation :

Finally, our study was insufficient to demonstrate an effect of partial venous occlusion on the inflammatory marker, oxidative stress, and acid-base balance in the ischemic limb. However, this study was not designed or powered to look at the impact on those parameters, although those are likely of great interest. The study period may not have been long to detect significant changes.

6. The authors should additionally test the effect of the intentional partial occlusion of the CIV in porcine with chronic hindlimb ischemia, which is created by the CIA occlusion for several days. If impossible, the authors should not describe a clinical perspective regarding the CIV occlusion method for treating patients with critical limb ischemia.

A) Description regarding CLI were eliminated or modified in the introduction and discussion:

.The sentence ”Particularly in patients with CLI” was eliminated in the discussion.

.The whole paragraph was modified as below :

From “There are two fatal subgroups for PAD, critical limb ischemia (CLI) and acute limb ischemia (ALI). CLI is the most severe manifestation of PAD, being defined by ischemic rest pain, ulcers and/or gangrene and is usually caused by atherosclerosis in the lower extremity arteries. In contrast, ALI occurs when there is sudden decrease in limb perfusion that threatens limb viability and requires urgent treatment to prevent loss of limb [2]. Irrespective of the pathophysiological processes underlying the ischemia, the final results remain the same.” To “In particular, acute limb ischemia (ALI) occurs when there is sudden decrease in limb perfusion that threatens limb viability and requires urgent treatment to prevent loss of limb [2]. A lack of oxygen due to an interruption of the blood supply to an acutely occluded limb causes accumulation of various metabolites, production of reactive oxygen species (ROS) and an inflammatory reaction in conjugation with tissue swelling, threatening the possibility of limb loss and even death”

7. As for the statistical analysis of changes in the oxygen saturation following the intentional partial occlusion of the CIV, was the analysis adjusted by the baseline oxygen saturation before the occlusion?

A) adjusted 

Reviewer #2: The authors present a single institution study looking at the effect of intentional restriction of venous return on tissue oxygenation in a porcine model of acute limb ischemia and investigate whether partial venous occlusion in the common iliac vein (CIV) causes a further increase of venous oxygenation in a porcine model of acute hindlimb ischemia.

1.Just under 10 porcines were analyzed. While this is interesting work, I have some questions and suggestions for revisions. I think the number of the study is to small to draw conclusion, the authors should soften their statements

A) To address this reviewer’s comment, we added some sentences in the discussion section as below : 

Second, there are concern about the small number of experimental animals and a relatively short period of observation time. With the current amount of observations, however, there was obvious trend towards improvement of venous oxygen saturation over time, following intentional partial venous occlusion in the CIV. Possibly, the inclusion of an additional time point, e.g. 6 hours after the partial venous occlusion, may have revealed more information about the effect of intentional restriction of venous return on tissue oxygenation in the ischemic limb.

A) To soften the conclusion, final conclusion was modified as below :

Our results suggest potential positive therapeutic effect of partial venous occlusion to induce the enhancement of oxygen uptake in the ischemic limb tissue. The effect of partial venous occlusion on oxygenation in the ischemic limb may deserve to be investigated in patients for whom revascularization therapy is unsuccessful.

2. In Discussion part line 15 “An expectation for this may be that the oxygen consumption in the limb tissue is increased” do you mean expectation or explanation. I think you mean explanation.

A) Corrected (expectation->explanation)

3. The pathophysiological reasons of venous occlusion and increased oxygen saturation is not becoming clear. Please discuss it better. How do you explain this in the peripheral system?

A) To address the comment, we added a paragraph in the discussion, with adding the New figure 5 :

Partial venous occlusion in the CIV induces local stasis of highly oxygenated blood in the ischemic area. It causes a slight venous pressure elevation in the vein, and the sustaining high oxygen level would be helpful for oxygen uptake by ischemic limb tissue, restoration of cell metabolism and limb perfusion (Fig 5). 

4.We know in the peripheral system a venous congestion results in trophic skin disorder and venus ulcera and is contraindicated for wound healing.

A) To address the reviewer’s comment, some explanation were inserted in the discussion as below : 

On the other hand, there is concern that venous ulcer or venous stasis related life threating condition may occur due to venous congestion via the intentional partial venous occlusion causing pressure in the veins to increase. However, those catastrophic events are unlikely to develop because the intentional partial venous occlusion would be applied, temporarily.

5.How do you measure the saturation, this should be clarified.

A) A paragraph regarding gas analysis was inserted in the method as below : 

Blood gas analysis was performed to measure oxygen saturation, carbon dioxide, bicarbonate, and acid-base balance. A proper blood sample for blood gas analysis consists of a 2 to 3 ml blood specimen collected anaerobically in 3 ml plastic syringe fitted with a small bore needle. Heparin was added to the syringe as an anticoagulant. Any air bubbles inadvertently introduced during sampling was promptly evacuated. 

6. Page 3 line 21-23 : The authors try to transferr the pathophysiological process in the heart to the peripheral vessels. I think this need to be discussed much deeper. Arterial occlusion combined with venous occlusion results in the peripheral system to the situation phlegmasia coerlia dolens, which is a life threating condition. Who was lactate, do you measure any labor parameters.

A) Some paragraph supporting our hypothesis and answering the reviewer’s comment were inserted as below : 

There is concern that venous ulcer or venous stasis related life threating condition may occur due to venous congestion via the intentional partial venous occlusion causing pressure in the veins to increase. However, those catastrophic events are unlikely to develop because the intentional partial venous occlusion would be applied, temporarily.

7. PLOS authors have the option to publish the peer review history of their article (what does this mean?). If published, this will include your full peer review and any attached files. While revising your submission, please upload your figure files to the Preflight Analysis and Conversion Engine (PACE) digital diagnostic tool, https://pacev2.apexcovantage.com/. PACE helps ensure that figures meet PLOS requirements. To use PACE, you must first register as a user. Registration is free. Then, login and navigate to the UPLOAD tab, where you will find detailed instructions on how to use the tool. If you encounter any issues or have any questions when using PACE, please email PLOS at figures@plos.org. Please note that Supporting Information files do not need this step.

---

## [Decision Letter · Decision Letter 1]

14 Oct 2020

PONE-D-20-19360R1

Effect of intentional restriction of venous return on tissue oxygenation in a porcine model of acute limb ischemia

PLOS ONE

Dear Dr. Kim,

Thank you for submitting your manuscript to PLOS ONE. After careful consideration, we feel that it has merit but does not fully meet PLOS ONE’s publication criteria as it currently stands. Therefore, we invite you to submit a revised version of the manuscript that addresses the points raised during the review process.

We look forward to receiving your revised manuscript.

Kind regards,

Yoshihiro Fukumoto

Academic Editor

PLOS ONE

Reviewers' comments:

Reviewer's Responses to Questions

**Comments to the Author**

1. If the authors have adequately addressed your comments raised in a previous round of review and you feel that this manuscript is now acceptable for publication, you may indicate that here to bypass the “Comments to the Author” section, enter your conflict of interest statement in the “Confidential to Editor” section, and submit your "Accept" recommendation.

Reviewer #1: (No Response)

Reviewer #2: All comments have been addressed

2. Is the manuscript technically sound, and do the data support the conclusions?

Reviewer #1: Partly

Reviewer #2: Yes

3. Has the statistical analysis been performed appropriately and rigorously? 

Reviewer #1: I Don't Know

Reviewer #2: Yes

4. Have the authors made all data underlying the findings in their manuscript fully available?

Reviewer #1: No

Reviewer #2: Yes

5. Is the manuscript presented in an intelligible fashion and written in standard English?

Reviewer #1: Yes

Reviewer #2: Yes

6. Review Comments to the Author

Reviewer #1: First of all, the authors’ response comments and explanations for the reviewer’s questions have been unkind for the reviewer. In order to accept the author’s explanation for the second comment of the reviewer, it will be necessary some experiment to show numerous collaterals in the porcine limbs with acute ischemia. If impossible, the authors should describe the concern in the section of Discussion. Next, according to the authors’ seventh response, it seems that the authors have (had already?) performed some statistical analyses adjusted by the baseline data for the second (first?) submission of this paper. The authors should describe that in the section of Materials and Methods.

Reviewer #2: No comments, the authors have adequately addressed my comments raised in a previous round.

thanks.

7. PLOS authors have the option to publish the peer review history of their article (what does this mean?). If published, this will include your full peer review and any attached files.

Reviewer #1: No

Reviewer #2: **Yes: **Dr. N. Schahab

---

## [Author Response · Author response to Decision Letter 1]

22 Oct 2020

Review Comments to the Author

Reviewer #1: First of all, the authors’ response comments and explanations for the reviewer’s questions have been unkind for the reviewer. In order to accept the author’s explanation for the second comment of the reviewer, it will be necessary some experiment to show numerous collaterals in the porcine limbs with acute ischemia. If impossible, the authors should describe the concern in the section of Discussion. Next, according to the authors’ seventh response, it seems that the authors have (had already?) performed some statistical analyses adjusted by the baseline data for the second (first?) submission of this paper. The authors should describe that in the section of Materials and Methods: 

Prior to describe my answers, I would like to apologize for my unkindness in the previous answer letter (1st revision rebuttal) I've caused. There are two comments raised by the academic editor and reviewer.

1st comment answer : 

The background in PAD is extremely heterogenous with many factors involved, but the outstanding factor that limits the extent of ischemic limb damage following arterial occlusion is the degree of development of the collateral circulation. That is to say, development of collateral vasculature is key in compensating for arterial occlusion in the ischemic limb. Arterial anatomy of the porcine hind limb is different from that in human peripheral limb. We described (1) the different vascular collateral system in the pig, compared to that in human, and (2) the reason why the pig is likely to have a numerous collateral vessels in the setting of ALI model, as below : 

In addition, arterial anatomy of the porcine hind limb is different from that in human peripheral limb. In the porcine vascular system, the distal abdominal aorta generally trifurcates into the right and left iliac arteries, and the common internal iliac trunk which then bifurcates into the right and left internal iliac arteries [24]. Development of collateral vasculature via those internal iliac arteries might be one of the important mechanism in compensating for arterial occlusion in the porcine limb ischemic model. Our limb ischemic model was induced with arterial embolization in the CIA, not at the level of distal abdominal aorta. Thus, the patent common internal iliac trunk might be facilitate the development of collateral blood flow into the ischemic limb.

From the reference 24 (Gao Y, Aravind S, Patel NS, Fuglestad MA, Ungar JS, Mietus CJ, et al. Collateral Development and Arteriogenesis in Hindlimbs of Swine After Ligation of Arterial Inflow. J Surg Res 2020;249:168-79. https://doi: 10.1016/j.jss.2019.12.005. PMID: 31986359).

2nd comment answer) 

We have only 7 experimental pigs in this study, and there are 4 estimated variables (T0, T1, T2, T3) prior to T4 (the time just prior to the occlusion of venous occlusion). Therefore, we confess that adjustment of those data is difficult in my statistical ability. Nevertheless, all pigs have a very similar baseline characteristics from the beginning of the experiment, including their size, weight, height, and a baseline arterial (T0 in artery; mean value of 100% O2 saturation without any standard deviation) and venous oxygenation (T0 in vein; mean value of 85.57 ± 1.72 % O2 saturation, just-small 1.72 standard deviation). We believe that our data is a relatively homogenous. Instead, to intensify the statistical significance of our result and compensate the un-adjusted data, we made a new paragraph and figure regarding the relationship between hypoxic damage by arterial embolization (defined as T4 minus T0 in vein saturation) and treatment effect with venous occlusion (defined as T7 minus T4 in vein saturation) by performing a Spearman correlation as below: 

The sentence “Hypoxic damage by arterial embolization was defined as T4 minus T0 in vein saturation, and treatment effect with partial venous occlusion was defined as T7 minus T4 in vein saturation, respectively” was inserted in the method section.

A new figure (Fig 6) was created to compensate the unadjusted data in this study, strengthened the statistical power, and help readers to understand the study’s result more easily :

The sentence “Figure 6 shows that hypoxic damage by arterial embolization is positively associated with treatment effect with venous partial occlusion (r = 0.788, P-value=0.035). This strong positive correlation might be interpreted that the more severe hypoxic damage, the greater would be the treatment effect with partial venous occlusion.” was added in the discussion. 

Fig 6. Spearman correlation was performed for bivariate correlations between hypoxic damage by arterial embolization (defined as T4 minus T0 in vein saturation) and treatment effect with partial venous occlusion (defined as T7 minus T4 in vein saturation) to establish a relationship. 

Reviewer #2: No comments, the authors have adequately addressed my comments raised in a previous round.

---

## [Decision Letter · Decision Letter 2]

5 Nov 2020

PONE-D-20-19360R2

Effect of intentional restriction of venous return on tissue oxygenation in a porcine model of acute limb ischemia

PLOS ONE

Dear Dr. Kim,

Thank you for submitting your manuscript to PLOS ONE. After careful consideration, we feel that it has merit but does not fully meet PLOS ONE’s publication criteria as it currently stands. Therefore, we invite you to submit a revised version of the manuscript that addresses the points raised during the review process.

We look forward to receiving your revised manuscript.

Kind regards,

Yoshihiro Fukumoto

Academic Editor

PLOS ONE

Reviewers' comments:

Reviewer's Responses to Questions

**Comments to the Author**

1. If the authors have adequately addressed your comments raised in a previous round of review and you feel that this manuscript is now acceptable for publication, you may indicate that here to bypass the “Comments to the Author” section, enter your conflict of interest statement in the “Confidential to Editor” section, and submit your "Accept" recommendation.

Reviewer #1: (No Response)

2. Is the manuscript technically sound, and do the data support the conclusions?

Reviewer #1: Partly

3. Has the statistical analysis been performed appropriately and rigorously? 

Reviewer #1: I Don't Know

4. Have the authors made all data underlying the findings in their manuscript fully available?

Reviewer #1: Yes

5. Is the manuscript presented in an intelligible fashion and written in standard English?

Reviewer #1: Yes

6. Review Comments to the Author

Reviewer #1: Thank you for your second response to my first review comments. I have gotten two answer comments and a revised manuscript based on the answer comments from you; however, there are two problems to be resolved in the revised manuscript.

1. If the authors cannot perform some experiments to show numerous collaterals in the porcine limbs with acute ischemia, the authors should describe that in the section of Discussion in the revised manuscript as follows: “...might facilitate the development of collateral blood flow into the ischemic limbs. However, we did not actually show the collateral arteries by some experiments, such as angiography and histological staining of the ischemic tissues, in this study.”

2. As to the second answer comment about the statistical analyses, it is necessary to be checked by a statistician whether the analyses were correctly performed to show the significance of their experimental results.

7. PLOS authors have the option to publish the peer review history of their article (what does this mean?). If published, this will include your full peer review and any attached files.

Reviewer #1: No

---

## [Author Response · Author response to Decision Letter 2]

11 Nov 2020

Reviewer #1: Thank you for your second response to my first review comments. I have gotten two answer comments and a revised manuscript based on the answer comments from you; however, there are two problems to be resolved in the revised manuscript.

1) If the authors cannot perform some experiments to show numerous collaterals in the porcine limbs with acute ischemia, the authors should describe that in the section of Discussion in the revised manuscript as follows: “...might facilitate the development of collateral blood flow into the ischemic limbs. However, we did not actually show the collateral arteries by some experiments, such as angiography and histological staining of the ischemic tissues, in this study.”

Answer) We thank you for your precious comments of our experiment. We added and modified some sentences to reflect your opinions as follows : 

 However, we did not actually show the limb perfusion through collaterals, which could be assessed with peripheral angiography, non-invasive imaging modalities, or histological staining of the ischemic tissues. Swine models fulfilling all ALI characteristics, with evaluating the adequacy of limb perfusion through collaterals intraprocedurally, seem to be lacking in the scientific literature. When it comes to further researches following this study, a major challenge may be having technologies available that can show numerous collaterals in the ischemic limb.

2) As to the second answer comment about the statistical analyses, it is necessary to be checked by a statistician whether the analyses were correctly performed to show the significance of their experimental results.

Answer) Actually, we consulted with a statistical specialist, when we prepared for the proper and best answers of the review’s previous comments. He (Dr. Byoung Geol Choi, PhD in Korea University Guro hospital, e-mail address: trv940@naver.com) suggested one statistical technique-Pearson correlation coefficient, instead of statistical adjustment, because of a relatively small sample number in our study. At the present time, no one has been addressed our topic-partial venous occlusion in the ischemic limb.

 If we were to perform statistical adjustment with our data, two methods-multiple linear regression or ANCOVA might be used. We (with Dr Choi) did those two methods (but, it didn’t work well). Calculating a sufficient sample size in number, based on our study’s result, we need at least 30 pigs to be sacrificed more (the minimal number for statistical adjustment, including pig’s death rate during the procedure). Even in the published researches regarding coronary sinus reduction in swine (the study which motivated us to start our topic), as we know, experimental pigs were small in number. And, all pig have a similar baseline characteristics, including age, sex weight, and nutrition, etc. We fully understand what the reviewer recommends. On the other hand, this is the first pilot study in literature (there is no previous or any article on this topic). At first, we just wanted to observe if partial venous occlusion in the common iliac vein (CIV) might cause a further increase of venous oxygenation. In the next study, we can make a special venous occluder to be deployed in vivo and have a plan to do an animal study with a large number of pigs, enough to do any statistical methods. 

For all statistical methods used in our study, we consulted with Dr Byoung Geol Choi, PhD in Korea University Guro hospital. He have been participating in numerous projects, and published many clinical journals (e-mail address: trv940@naver.com). In our data, Pearson correlation coefficient is the most appropriate statistical method that he recommended. We will try to do any statistical analysis, if the reviewer suggests a specific statistical methods for adjustment.

---

## [Decision Letter · Decision Letter 3]

16 Nov 2020

Effect of intentional restriction of venous return on tissue oxygenation in a porcine model of acute limb ischemia

PONE-D-20-19360R3

Dear Dr. Kim,

We’re pleased to inform you that your manuscript has been judged scientifically suitable for publication and will be formally accepted for publication once it meets all outstanding technical requirements.

Kind regards,

Yoshihiro Fukumoto

Academic Editor

PLOS ONE

Additional Editor Comments (optional):

Reviewers' comments:

Reviewer's Responses to Questions

**Comments to the Author**

1. If the authors have adequately addressed your comments raised in a previous round of review and you feel that this manuscript is now acceptable for publication, you may indicate that here to bypass the “Comments to the Author” section, enter your conflict of interest statement in the “Confidential to Editor” section, and submit your "Accept" recommendation.

Reviewer #1: All comments have been addressed

2. Is the manuscript technically sound, and do the data support the conclusions?

Reviewer #1: Yes

3. Has the statistical analysis been performed appropriately and rigorously? 

Reviewer #1: Yes

4. Have the authors made all data underlying the findings in their manuscript fully available?

Reviewer #1: Yes

5. Is the manuscript presented in an intelligible fashion and written in standard English?

Reviewer #1: Yes

6. Review Comments to the Author

Reviewer #1: The authors have adequately addressed my comments raised in the previous reviews. I have no more comments. Thank you.

7. PLOS authors have the option to publish the peer review history of their article (what does this mean?). If published, this will include your full peer review and any attached files.

Reviewer #1: No

---

## [Editor Report · Acceptance letter]

2 Dec 2020

PONE-D-20-19360R3 

Effect of intentional restriction of venous return on tissue oxygenation in a porcine model of acute limb ischemia 

Dear Dr. Kim:

I'm pleased to inform you that your manuscript has been deemed suitable for publication in PLOS ONE. Congratulations! Your manuscript is now with our production department. 

Kind regards, 

on behalf of

Dr. Yoshihiro Fukumoto 

Academic Editor

PLOS ONE